# 2DiNTS: 2D Differentiable Neural Network Topology Search for Multi-Modal Cell Segmentation

**Hussam Azzuni**
Department of Computer Vision
Mohamed bin Zayed University of Artificial Intelligence
Abu Dhabi, UAE
hussam.azzuni@mbzuai.ac.ae

**Muhammad Hamza Sharif**
Mohamed bin Zayed University of Artificial Intelligence
Abu Dhabi, UAE
muhammad.sharif@mbzuai.ac.ae

**Naveed Syed**
Sheikh Shakhbout Medical City
Abu Dhabi, UAE
nasyed@ssmc.ae

**Mohammad Yaqub**
Mohamed bin Zayed University of Artificial Intelligence
Abu Dhabi, UAE
mohammad.yaqub@mbzuai.ac.ae

**Min Xu**
Mohamed bin Zayed University of Artificial Intelligence
Abu Dhabi, UAE
Min.Xu@mbzuai.ac.ae

## Abstract

Cell segmentation is a crucial step in various biological and biomedical applications. However, microscopic images could vary depending on the lighting conditions employed. Training a deep learning model to handle various modalities could prove quite difficult since it is much harder to create a comprehensive model. Our proposed method tackles four different microscopic imaging modalities such as brightfield, fluorescent, phase-contrast, and differential interference contrast to create a flexible multi-modal cell segmentation model. Differentiable Neural Network Topology Search (DiNTS) helps in providing a search space to accommodate the different features between the four modalities, which resulted in an increase of 0.5% F1 on the validation set.

## 1 Introduction

Cell segmentation in microscopy images is often the first step in the quantitative analysis of imaging data for biological and biomedical applications. Microscopy allows capturing structural and functional properties of biological model systems, including cultures of cells, tissues, and organoids. The advancement of microscopy for capturing such systems in greater detail reveals new insights about understanding the complex biological patterns in living organisms. To extract meaningful information from the imaging analysis using microscopy, the primary task is to identify (segment) the cell nuclei which is used to count cells, track populations, locate proteins, and classify phenotypes or profile treatments. Different approaches are used to identify nuclei using classical segmentation algorithms

36th Conference on Neural Information Processing Systems (NeurIPS 2022).

such as thresholding, watershed or active contours. However, these traditional approaches are not suitable for segmenting cell in all types of microscopy images due to differences of microscopy modalities, scales and experimental conditions and requires the need of deep learning based method that extracts the features from various modalities. In general, different microscopy techniques are used in research to capture biological patterns which are mainly divided into light and electron microscopy due to their mechanism of capturing cellular images. Light microscopy, uses beams of visible light to capture color images and it has less resolution and magnification power than electron microscopy. As electron microscopic images have higher resolution, such images captured by this technique are highly recommended for studying the 3D cellular structure at the atomic level. But the use of light microscopy is always the optimum method for segmenting cells, whether by manual effort or a computational approach. As implied by its name, light microscopy is used to capture cellular structure in the presence of light. As visible light has different color, based on the different lighting condition used in different circumstance, light microscopic images are divided into different modalities which are named as: bright field, dark field, phase-contrast, differential interference contrast, fluorescent, and confocal. There are various challenges present in multi-modal cell segmentation. First, The deep learning model finds it considerably harder to generalize to the four modalities because of the visual differences between the various modalities. Second, each modality has its own unique artifacts, making it more difficult to customize solutions for each modality. Last but not least, overlapping cells make it exceedingly challenging to get the best instance segmentation masks.

The proposed architecture is a versatile algorithm to achieve optimal performance on all four modalities. The most effective and economical topology for multi-modal cell segmentation is thus found using Differentiable Neural Network Topology Search (DiNTS), which is employed to attain such flexibility.

# 2 Method

## 2.1 Preprocessing

Given the various modalities, it is important to ensure that all the images consists of the same number of channels. For this reason, if an image only has one channel, that channel would be repeated to provide a 3-channel input. Then, all the images go through a normalization process.

## 2.2 2DiNTS

Differentiable Neural Network Topology Search (DiNTS) [1] was originally designed for 3D medical image segmentation. It utilizes a differentiable search algorithm to find the optimal network, based on the number of layers, the neurons of each layer and the cell operation used. For our method, DiNTS was adapted for 2D Multi-Modal cell segmentation. Monai implementation [2] of DiNTS has upsampling, downsampling, and two cell operations such as skip connection, or a 3x3 convolution. Since there was initially two cell operations, four other cell operations were added to increase the search space to eventually improve the overall performance, leading to the following cell operations:

1. Skip connection

2. 1x1 convolution

3. 3x3 convolution

4. Depthwise 3x3 convolution

5. 5x5 convolution

6. 7x7 convolution

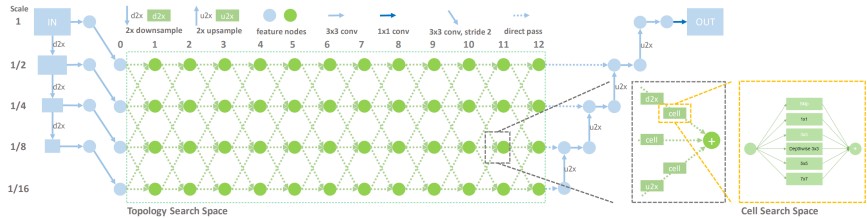

Figure 1: Redesigned for 2D cell segmentation [1] showing the additional cell operations in black in the cell search block.

Table 1: Development environment and requirements.

| System | Ubuntu 20.04.4 LTS |
|---|---|
| GPU (number and type) | One Quadro RTX 6000 |
| CUDA version | 11.0 |
| Programming language | Python 3.6.13 |
| Deep learning framework | Pytorch (Torch 1.10.1, torchvision 0.11.2) |

## 3 Experiments

### 3.1 Dataset

The dataset contains 1000 labeled images from 4 modalities such as Brightfield (300), Fluorescent (300), Phase-contrast (200), Differential interference contrast (200). These images are from different tissue, staining techniques, and different resolutions.

### 3.2 Implementation Details

Various implementations were tested such as increasing the number of blocks inside the search space, cell operations, and the resolution. The model with the best performance is saved and used for inference after the search is complete. Additionally, Multiple baselines such as UNet and SegResNet were examined to confirm the viability of 2DiNTS. The development environment and requirements are presented in Table 1. The training protocol used for all models is shown in table 2. Both training and inference used the same patch size.

## 4 Results and Discussion

Table 4 shows the performance comparison between UNet, SegResNet, and DiNTS on Semantic segmentation. This could be contributed to the adaptive nature of DiNTS. However, this comes at the cost of a huge training time as shown in table 3. Opposite to my original hypothesis, increasing the number of operations resulted in a lower Dice score to the original implementation of only 2 operations. One explanation is that having so many options for operations may eventually cause people to select less-than-ideal cell procedures.

Table 2: Training protocol.

| **Network initialization** | Random Initialization |
|---|---|
| **Training:Validation samples** | 900:100 images |
| **Total epochs** | 2000 |
| **Optimizer** | AdamW |
| **Initial learning rate (lr)** | 6e-4 |
| **Early stopping** | Patience = 100 |
| **Loss function** | DiceCELoss [3] |
| **Patch size** | 224x224 **or** 384x384 |

Table 3: Parameters and Training time for each model. All models were trained on 224x224, while *
indicates training on 384x384. All DiNTS models were trained on the original two operations, whilst
** indicates the usage of six 2D operations.

| Architecture | Training time | Parameters |
|---|---|---|
| UNet | ≈ 11 hours and 45 minutes | 1,626,072 |
| SegResNet (Pixel shuffle) | ≈ 7 hours and 46 minutes | 443,747 |
| SegResNet (Non-trainable) | ≈ 9 hours and 11 minutes | 395,139 |
| SegResNet (Deconv) | ≈ 17 hours and 35 minutes | 400,571 |
| DiNTS (Mul = 0.5) (Depth = 8) | ≈ 16 hours and 32 minutes | 3,782,259 |
| DiNTS (Mul = 1.0) (Depth = 8) | ≈ 21 hours and 50 minutes | 15,115,251 |
| DiNTS (Mul = 1.0) (Depth = 8)* | ≈ 34 hours and 5 minutes | 15,115,251 |
| DiNTS (Mul = 1.0) (Depth = 8)** | ≈ 109 hours and 12 minutes | 15,115,251 |
| DiNTS (Mul = 1.0) (Depth = 12) | ≈ 41 hours and 42 minutes | 22,463,595 |

Table 4: Validation results.

| Model | Resolution | Batch size | Dice score |
|---|---|---|---|
| UNet | 256×256 | 16 | 0.726 |
| SegResNet (Non-trainable upsampling) | 256×256 | 16 | 0.722 |
| SegResNet (Deconv upsampling) | 256×256 | 16 | 0.736 |
| SegResNet (Pixel-Shuffle upsampling) | 256×256 | 16 | 0.722 |
| DiNTS (Mul 0.5) (Depth 8) | 256×256 | 16 | 0.746 |
| DiNTS (Mul 1) (Depth 8) | 256×256 | 8 | 0.749 |
| DiNTS (Mul 1) (Depth 8) (6 ops) | 256×256 | 8 | 0.742 |
| DiNTS (Mul 1) (Depth 8) | 384×384 | 4 | **0.755** |
| DiNTS (Mul 1) [Depth 12] | 256×256 | 6 | 0.753 |

## 4.1 Quantitative Results on Tuning Set

Table 5 shows the instance segmentation performance on F1 score. The huge drop in performance
could be allocated to the difference between semantic and instance segmentation. Even though our
2D DiNTS model could perform well in semantic segmentation, this does not mean it is optimal for
instance segmentation; due to the fact that the semantic segmentation results are overlapping resulting
in poor instance differentiation.

## 4.2 Limitation and future work

One of the biggest limitations in the proposed method is the presence of clustered nuclei. 2DiNTS
uses 2D operations to find the most optimal topology to eventually obtain a final segmentation
mask, but these improvements does not necessarily reflect on instance segmentation as it is abundant
with clustered nuclei. A possible direction is to utilize SOTA method for cell segmentation such as
Omnipose [4] to carefully design a search space that is more customized towards touching cells.

Table 5: Results based on the Fine Tuning set

| Model | Resolution | F1 score |
|---|---|---|
| SegResNet (Deconv upsampling) | 256x256 | 0.540 |
| DiNTS (Mul 1) (Depth 12) | 256x256 | 0.549 |
| DiNTS (Mul 1) (Depth 8) | 384x384 | **0.553** |

# 5    Conclusion

In this work, we propose the usage of DiNTS to obtain a flexible algorithm to accommodate the variety between the four modalities. This resulted in an improvement of 2.9% DSC on the validation set compared to Vanilla U-Net. However, this does not reflect on instance segmentation performance; due to the over-segmentation present resulting in poorer instance predictions.

## Acknowledgement

The authors of this paper declare that the segmentation method they implemented for participation in the NeurIPS 2022 Cell Segmentation challenge has not used any private datasets other than those provided by the organizers and the official external datasets and pretrained models. The proposed solution is fully automatic without any manual intervention.

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
