# OpenReview forum: "2DiNTS: 2D Differentiable Neural Network Topology Search for Multi-Modal Cell Segmentation"
_NeurIPS.cc/2022/Challenge/CellSeg — Submitted to NeurIPS CellSeg 2022_

### Official Review · Reviewer_45p1 · 2023-01-15
**Cell Segmentation with 2D DiNTS**

**Rating:** 5
**Confidence:** 4

**Review:**

The authors tried to apply 2D DiNTS to achieve a search space and accommodate different features between four modalities. The overall method is good but without any obvious novelty.


The overall writing is great and well-structured. Some parts of the presentation are not clear in some cases. Before accepting the paper I would suggest addressing these points.

1. Figure 1 was given but there was no corresponding explanation about the detailed network in the main content. How does the topology search work in the presented network?

2. Figure 1 was redesigned from Figure 2 in the paper [1]. What is the redesigned part? Is the cell search space?

3. In Section 3.2, additional baselines were examined to confirm the viability of 2DiNTS. Which baselines were not clearly mentioned.

4. The loss function was not discussed in the paper.

5. As the authors discussed in the limitation section, 2D DiNTS didn’t consider high-level semantic information. It’s a possible reason but there were not sufficient experiments in the paper. I would suggest authors re-thinking the search space. (The current model performance is just ok, not outstanding. The original 3D DiNTS compared with nnU-Net. Did the authors consider nnU-Net?)

6. Some obvious typos:

(1) On Page 2, by the end of the Introduction, *the* deep learning model finds it considerably harder to generalize to the four modalities because of the visual differences between the various modalities.

(2) On Page 2, in the Method section, Since this method only contains two cell operations*, various* other cell operations were added.

(3) On Page 3, in the Implementation section, two additional?

---

### Official Review · Program_Chairs · 2023-01-15
**poor writing and performance**

**Rating:** 4
**Confidence:** 5

**Review:**

The author didn't give a detailed description of the method and the validation performance is poor. It doesn't worth carefully reviewing.

---

### Decision · Program_Chairs · 2023-01-19

Reject